# Activating the regenerative potential of Müller glia cells in a regeneration-deficient retina

**Katharina Lust[1,2], Joachim Wittbrodt[1]\***

[1]Centre for Organismal Studies, Heidelberg University, Heidelberg, Germany;
[2]Hartmut Hoffmann-Berling International Graduate School, Heidelberg, Germany

**Abstract** Regeneration responses in animals are widespread across phyla. To identify molecular players that confer regenerative capacities to non-regenerative species is of key relevance for basic research and translational approaches. Here, we report a differential response in retinal regeneration between medaka (*Oryzias latipes*) and zebrafish (*Danio rerio*). In contrast to zebrafish, medaka Müller glia (olMG) cells behave like progenitors and exhibit a restricted capacity to regenerate the retina. After injury, olMG cells proliferate but fail to self-renew and ultimately only restore photoreceptors. In our injury paradigm, we observed that in contrast to zebrafish, proliferating olMG cells do not maintain *sox2* expression. Sustained *sox2* expression in olMG cells confers regenerative responses similar to those of zebrafish MG (drMG) cells. We show that a single, cell-autonomous factor reprograms olMG cells and establishes a regeneration-like mode. Our results position medaka as an attractive model to delineate key regeneration factors with translational potential.

DOI: https://doi.org/10.7554/eLife.32319.001

**\*For correspondence:**
jochen.wittbrodt@cos.uni-heidelberg.de

**Competing interests:** The authors declare that no competing interests exist.

## Introduction

The ability to regenerate individual cells, lost organs or even the structure of the entire body is widespread in the animal kingdom. The means by which certain species achieve remarkable feats of regeneration, whereas others have restricted or no capacity to do so is poorly understood. Teleost fishes are widely used models to study development, growth and regeneration of the visual system (*Centanin et al., 2011*; *Raymond et al., 1988*, *2006*; *Rembold et al., 2006*). The retina of these fish undergoes lifelong neurogenesis, and the range of retinal cell types is generated from two sources. The first are the cells of the ciliary marginal zone (CMZ), which include retinal stem cells that give rise to progenitor cells and ultimately differentiated cell types of the growing neural retina (*Centanin et al., 2011*, *2014*; *Raymond et al., 2006*). A second source for new retinal cells are Müller glia (MG) cells, which generate new cell types during homeostasis and regeneration (*Bernardos et al., 2007*).

Some teleost species, including goldfish (*Carassius auratus*) and zebrafish (*Danio rerio*), have been analyzed with respect to their ability to regenerate the retina and recover visual function after injuries (*Bernardos et al., 2007*; *Braisted and Raymond, 1992*; *Raymond et al., 1988*; *Sherpa et al., 2008*). Among these, zebrafish is the best-studied and has been shown to contain multipotent MG cells which can self-renew and regenerate all retinal neuronal and glial cell types after injuries. It is currently assumed that other teleost species possess the same regenerative capacities; however, detailed analyses have been lacking.

To investigate MG cell-mediated retina regeneration in a distantly related teleost, we chose the Japanese ricefish medaka (*Oryzias latipes*), which is a well-established model organism that shared its last common ancestor with zebrafish between 200 and 300 million years ago (*Schartl et al.,*

**eLife digest** All animals have at least some ability to repair their bodies after injury. But certain species can regenerate entire body parts and even internal organs. Salamanders, for example, can regrow their tail and limbs, as well as their eyes and heart. Many species of fish can also regenerate organs and tissues. In comparison, mammals have only limited regenerative capacity. Why does regeneration vary between species, and is it possible to convert a non-regenerating system into a regenerating one?

Laboratory studies of regeneration often use the model organism, zebrafish. Zebrafish can restore their sight after an eye injury by regenerating the retina, the light-sensitive tissue at the back of the eye. They are able to do this thanks to cells in the retina called Müller glial cells. These behave like stem cells. They divide to produce identical copies of themselves, which then transform into all of the different cell types necessary to produce a new retina.

Lust and Wittbrodt now show that a distant relative of the zebrafish, the Japanese ricefish 'medaka', lacks these regenerative skills. Although Müller glial cells in medaka also divide after injury, they give rise to only a single type of retinal cell. This means that these fish cannot regenerate an entire retina. Lust and Wittbrodt demonstrate that in medaka, but not zebrafish, levels of a protein called Sox2 fall after eye injury. As Sox2 has been shown to be important for regeneration in zebrafish Müller glial cells, the loss of Sox2 may be preventing regeneration in medaka. Consistent with this, restoring Sox2 levels in medaka Müller glial cells enabled them to turn into several different types of retinal cell.

Sox2 is also present in the Müller glial cells of other species with backbones, including chickens, mice, and humans. Future experiments should test whether loss of Sox2 after injury contributes to the lack of regeneration in these species. If it does, the next question will be whether restoring Sox2 can drive a regenerative response.

DOI: https://doi.org/10.7554/eLife.32319.002

*2013*). Few regeneration studies have been carried out in medaka, but the literature reveals some interesting differences to zebrafish. Whereas fins can be fully regenerated in adult medaka (*Nakatani et al., 2007*), the heart has no regenerative capacity (*Ito et al., 2014*; *Lai et al., 2017*). The development and growth of the neural retina of medaka has been studied (*Centanin et al., 2011*, *2014*; *Martinez-Morales et al., 2009*), but regeneration studies are missing.

After injuries, multipotent MG cells of the zebrafish retina have been shown to upregulate the expression of pluripotency factors including *lin-28*, *oct-4*, *c-myc* and *sox2* (*Ramachandran et al., 2010*). Sox2 is well known for its role in maintaining the pluripotency of embryonic stem cells (*Masui et al., 2007*) and is one of the four original Yamanaka factors required for the generation of induced pluripotent stem cells (*Takahashi et al., 2007*). Sox2 has been frequently used in reprogramming studies, such as the conversion of mouse and human fibroblasts directly into induced neural stem cells (*Ring et al., 2012*), or the transformation of NG2 glia into functional neurons following stab lesions in the adult mouse cerebral cortex (*Heinrich et al., 2014*). In the regenerating zebrafish retina, *sox2* expression is upregulated 2 days post injury (dpi) and is necessary and sufficient for the MG proliferation associated with regeneration (*Ramachandran et al., 2010*; *Gorsuch et al., 2017*).

In the present study, we find that medaka MG (olMG) cells display a restricted regenerative potential after injury and only generate photoreceptors (PRCs). We observed that olMG cells can re-enter the cell cycle after injures but fail to divide asymmetrically or generate neurogenic clusters, two steps which are essential to full regeneration. Using in vivo imaging, two-photon mediated specific cell ablations and lineage tracing, we find that olMG cells react preferentially to injuries of PRCs and are only able to regenerate this cell type. We demonstrate that *sox2* is expressed in olMG cells in the absence of injury but, in contrast to zebrafish, is not maintained in proliferating olMG cells after injury. We show that inducing targeted expression of *sox2* in olMG cells is sufficient to shift olMG cells into a regenerative mode reminiscent of zebrafish, where they self-renew and regenerate multiple retinal cell types.

## Results

### olMG cells re-enter the cell cycle after injury but do not generate neurogenic clusters

In contrast to zebrafish and goldfish, where MG cells are described as the source of rod PRCs that gradually accumulate during the early larval period (*Bernardos et al., 2007*; *Nelson et al., 2008*), it has been shown previously that olMG cells are quiescent at a comparable developmental stage in the hatchling (8 dpf) retina (*Lust et al., 2016*). While the zebrafish retina massively increases its rod PRC number during post-embryonic growth (*Figure 1—figure supplement 1A–B'''*) via the proliferation of MG cells (*Bernardos et al., 2007*), the medaka retina maintains its rod PRC layer from embryonic to adult stages (*Figure 1—figure supplement 1C–D'''*) and rod PRCs are born from the CMZ (*Figure 1—figure supplement 2*).

In order to address the regenerative abilities of olMG cells we used the *rx2*::H2B-eGFP transgenic line that labels the CMZ, olMG cells and cone PRCs but no rods in hatchling (8dpf) and adult medaka (*Martinez-Morales et al., 2009*; *Reinhardt et al., 2015*) (*Figure 1—figure supplement 3*). To investigate the reaction of olMG cells and the retina upon injury, we performed needle injuries on *rx2*::H2B-eGFP transgenic fish. To label cells re-entering the cell cycle we subsequently analyzed the fish either by immunohistochemistry for the mitotic marker phospho-histone H3 (PH3) at 3 dpi or incubated them in BrdU for 3 days to label cells in S-phase. We detected proliferating cells in the central retina, on the basis of both labels PH3 (*Figure 1A–1A''*) and BrdU (*Figure 1B–1B''*) 3 days after a needle injury. These proliferating cells were also positive for *rx2*-driven H2B-eGFP, showing that the olMG cells had re-entered the cell cycle. These results demonstrate that olMG cells in hatchling medaka are quiescent in an uninjured background (*Lust et al., 2016*), but begin to proliferate upon injury.

The onset of MG proliferation in zebrafish has been observed between 1 and 2 dpi (*Fausett and Goldman, 2006*). To understand if olMG cells show a similar mode of activation, we performed BrdU incorporation experiments and analyzed time-points after injury ranging from 1 dpi until 3 dpi. At 1 dpi, no BrdU-positive cells were detected in the retina (data not shown). At 2 dpi, the first BrdU-positive cells were detected in the inner nuclear layer (INL) and the outer nuclear layer (ONL) of the central retina (*Figure 1—figure supplement 4A–B'''*). Co-localization with GFP showed that these cells are olMG cells or olMG-derived cells (*Figure 1—figure supplement 4A–B'''*).

In response to injury olMG cells initiate DNA synthesis and divide maximally once as indicated by the appearance of single or a maximum of two BrdU-positive cells next to each other in the INL at both 2 dpi and 3 dpi (*Figure 1C and C'*).

In contrast, the injury response of zebrafish MG (drMG) cells at comparable stages (4dpf) is characterized by the formation of large nuclear, neurogenic clusters in the INL (*Figure 1D and D'*). This is consistent with the response of adult drMG cells to injury in which a single asymmetric division produces a MG cell and a progenitor cell that divides rapidly to generate neurogenic clusters (*Nagashima et al., 2013*).

These results show that olMG cells start re-entering the cell cycle between 1 and 2 dpi but do not generate neurogenic clusters.

### olMG cells react preferentially to PRC injuries by apical migration

For proper regeneration to occur, the appropriate cell types must be produced. This requires not only the regulation of the proliferation of stem or progenitor cells, but also the proper control of lineage decisions in the progenitors. If and when fate decisions are made by the MG cells or proliferating progenitors during regeneration is largely unknown. To study whether different injury sites (PRC or retinal ganglion cell (RGC) injury) result in a differential response of olMG cells, we used two-photon mediated ablations and consecutive imaging (*Figure 2—figure supplement 1A–D*) and addressed their behavior in immediate (up to 30 hours post injury (hpi)) and late (until 6 dpi) response to injury.

We induced PRC injuries in medaka and observed that olMG nuclei below the wound site started migrating apically at 17 hpi (*Figure 2A–2A'''*, see also *Figure 2—video 1*). These migrations were not coordinated between individual cells. Some nuclei migrated into the ONL, whereas others stayed at the apical part of the INL. Nuclei farther from the wound site did not migrate in response

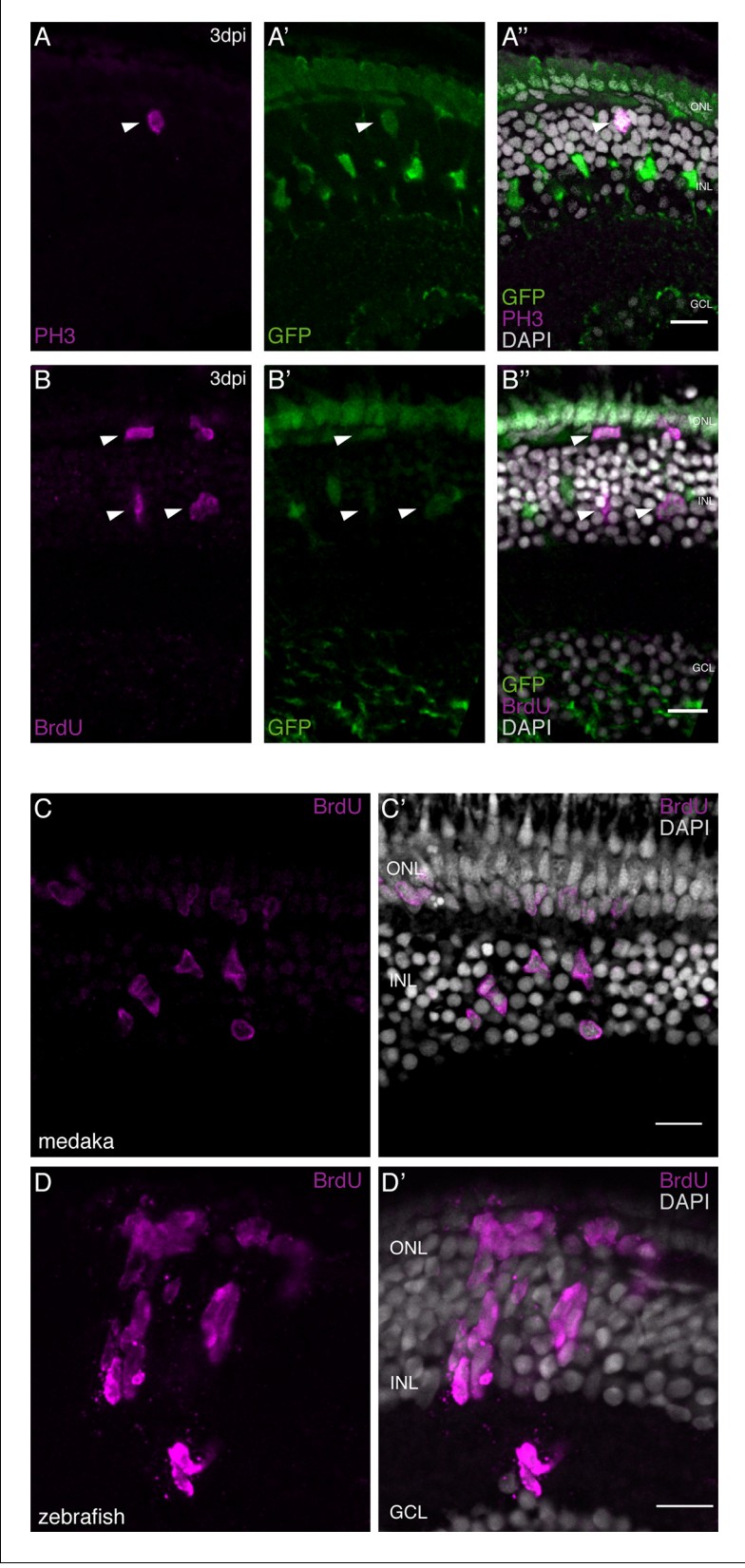

**Figure 1.** olMG cells re-enter the cell cycle after injury but do not generate neurogenic clusters.  (A–A'')
Cryosection of a needle-injured hatchling medaka retina of the transgenic line *rx2*::H2B-eGFP. PH3 stainings
*Figure 1 continued on next page*

*Figure 1 continued*

(magenta) on the hatchling medaka retinae 3 days post needle injury show mitotic cells present in the central retina (arrowhead), co-localizing with the *rx2* nuclear reporter expression (green). (n = 4 fish, data obtained from two independent experiments). (B–B'') Cryosection of a needle-injured hatchling medaka retina of the transgenic line *rx2*::H2B-eGFP. A 3-day pulse of BrdU marks proliferating cells in the central retina after needle injury (arrowheads). BrdU staining (magenta) co-localizes with *rx2* nuclear reporter expression (green), indicating that olMG cells re-entered the cell cycle. (n = 6 fish, data obtained from three independent experiments). (C, C') Cryosection of a needle-injured hatchling medaka retina. BrdU-positive (magenta) single cells are present in the INL and ONL. (n = 6 fish, data obtained from two independent experiments). (D, D') Cryosection of a needle-injured zebrafish retina. BrdU-positive (magenta) neurogenic clusters are present in the INL. Additionally, BrdU-positive proliferating cells can be detected in the ONL (n = 3 fish, data obtained from two independent experiments). Scale bars are 10 μm.

DOI: https://doi.org/10.7554/eLife.32319.003

The following figure supplements are available for figure 1:

**Figure supplement 1.** Rod photoreceptor density is increased during postembryonic growth of zebrafish but not medaka.
DOI: https://doi.org/10.7554/eLife.32319.004

**Figure supplement 2.** Rx2-positive CMZ cells generate rod photoreceptors, during post-embryonic growth.
DOI: https://doi.org/10.7554/eLife.32319.005

**Figure supplement 3.** *Rx2*-reporter labels olMG cells, cone PRCs and CMZ cells in the hatchling medaka retina.
DOI: https://doi.org/10.7554/eLife.32319.006

**Figure supplement 4.** Injury-induced timing of olMG cell cycle re-entry.
DOI: https://doi.org/10.7554/eLife.32319.007

to the injuries. In contrast, after RGC injuries, there was no migration of olMG nuclei, either apically or basally toward the wound, within the first 30 hpi (*Figure 2B–2B'''*, see also *Figure 2—video 1*).

To investigate whether olMG nuclei migrate back at later time-points after PRC injuries or show any migratory behavior after RGC injuries, we re-imaged the injury site at 2-day intervals to follow an injured retina up to 6 dpi. At 2 dpi, retinae with PRC injuries showed a gap in the INL below the injury site, at a position where olMG nuclei are normally found, reflecting the migration of olMG nuclei towards the ONL from this location (*Figure 2C–2C''*). The gap in the INL persisted until 6 dpi (*Figure 2C''*). The reaction of olMG cells in retinae with RGC injuries differed. Here, we neither observed an apical nor basal migration of olMG nuclei (*Figure 2D–2D''*) and in fact no migration of olMG nuclei was observed at all until 6 dpi. To rule out that this is due to too little damage in the RGC layer we increased the injury size. This led to swelling and secondary cell death of PRCs and activated olMG nuclei to migrate apically (*Figure 2—figure supplement 2A–B*), indicating further that their preferential reaction is toward PRC injuries.

Taken together, these results show that olMG nuclei migrate toward PRC injury sites within 24 hpi and remain in this location up until 6 days, whereas they display no discernible reaction toward RGC injuries. This indicates a clear preferential reaction of olMG nuclei to refill the injured PRC layer.

## olMG nuclei but not their cell bodies are depleted after PRC injuries

Long-term in vivo imaging of fish that were injured in the ONL made it apparent that olMG nuclei migrate apically into the wound site but remain there which might indicate a complete remodeling of the soma of these neuroepithelial cells. To understand whether cell bodies of the olMG cells remain intact during this nuclear migration, we analyzed nuclear movements (transgenic line *rx2*::H2B-eGFP) in the context of the olMG cell body (transgenic line *rx2*::lifeact-eGFP). We imaged the double transgenic animals at 2-day intervals following ONL injuries. As previously observed, olMG nuclei migrated out of the INL into the wound site (*Figure 3A–3A''*). The *rx2*::lifeact-eGFP labeled cell bodies of the olMG cells spanning the entire apico-basal distance remained intact until 6 dpi in the absence of an apparent (labeled) nucleus in the INL (*Figure 3A''*). The earlier position of the nucleus was still recognizable by a slight enlargement of the soma.

Additionally, to extend the range of analysis, we performed immunohistochemistry on injured fish at 3 and 10 dpi. After injury, incubation in BrdU for 3 days and direct fixation at 3 dpi we found that at the site of injury the GFP-positive cell bodies, labeled by *rx2*::lifeact-eGFP, did not contain a

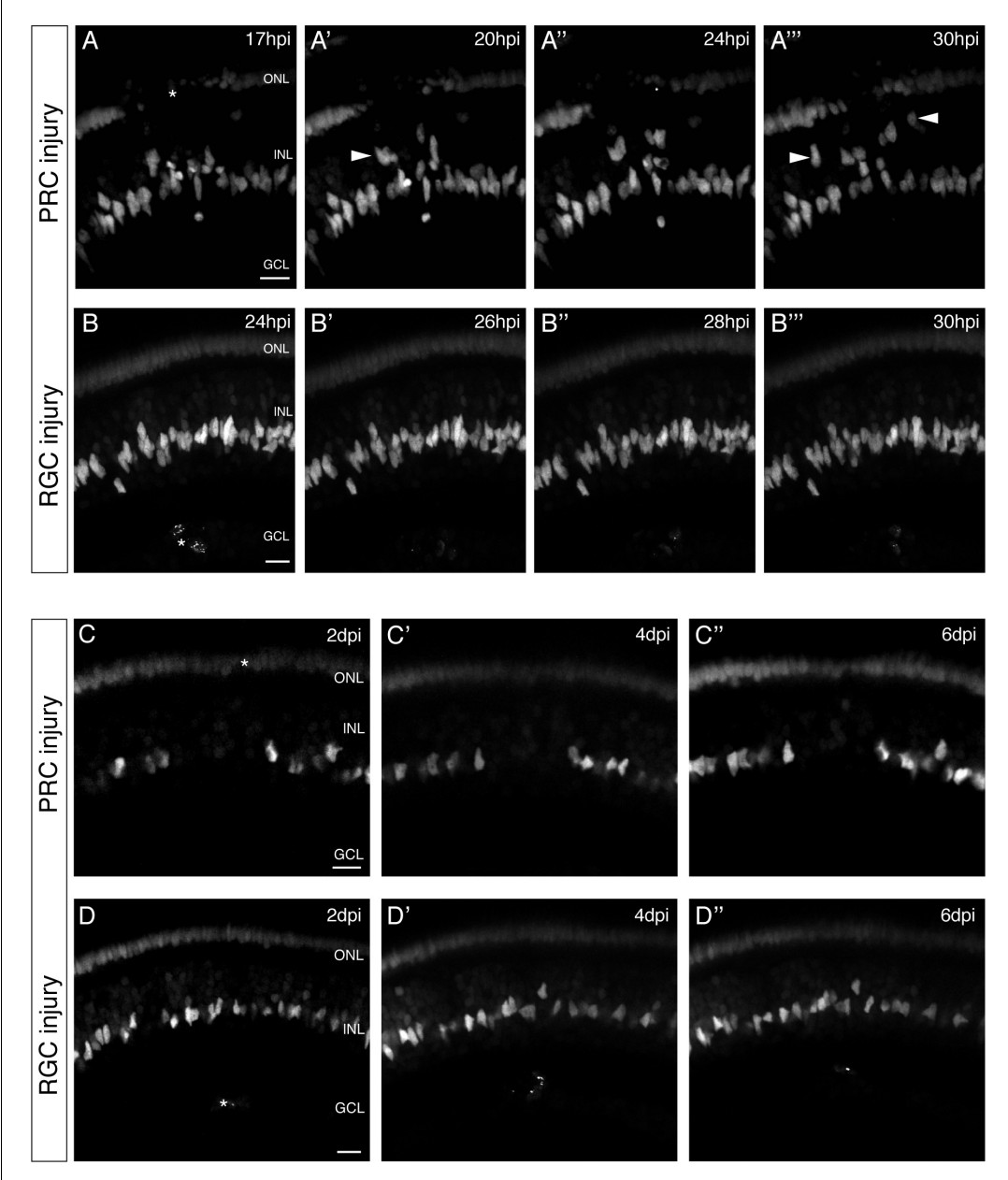

**Figure 2.** olMG cells react preferentially to PRC injuries by apical migration. (**A–B'''**) In vivo imaging of hatchling *rx2*::H2B-eGFP medaka retinae which were either injured in the ONL or the ganglion cell layer (GCL) (asterisks) using a two-photon laser and imaged consecutively until 30 hpi (n > 10 fish each, data obtained from >10 independent experiments each). (**A–A'''**) After PRC injuries olMG nuclei (arrowheads) start migrating apically towards the ONL layer from 17 hpi on. The migration is not coordinated among different migrating nuclei. (**B–B'''**) After RGC injuries no migration of olMG nuclei can be detected until 30 hpi. Scale bars are 10 µm. (**C–D''**) In vivo imaging of hatchling *rx2*::H2B-eGFP medaka retinae which were either injured in the ONL or the GCL (asterisks) using a two-photon laser and imaged every second day after injury (n > 10 fish each, data obtained from >10 independent experiments each). (**C–C''**) PRC injuries result in an apical migration of olMG nuclei into the injury site. The following days until 6 dpi the nuclei do not migrate back toward the INL resulting in a gap of olMG nuclei in the INL. (**D–D''**) After RGC injuries no migration of olMG nuclei can be detected until 6 dpi. Scale bars are 10 µm.

DOI: https://doi.org/10.7554/eLife.32319.008

The following video and figure supplements are available for figure 2:

**Figure supplement 1.** Two-photon mediated laser ablation enables targeted cell ablation in the retina resulting in specific cell death signatures.

DOI: https://doi.org/10.7554/eLife.32319.009

**Figure supplement 2.** Increased RGC injuries lead to swelling and secondary cell death in the PRC layer.

DOI: https://doi.org/10.7554/eLife.32319.010

*Figure 2 continued on next page*

*Figure 2 continued*
**Figure 2—video 1.** In vivo imaging of olMG nuclei reactions to a PRC injury
DOI: https://doi.org/10.7554/eLife.32319.011
**Figure 2—video 2.** In vivo imaging of olMG nuclei reactions to a RGC injury
DOI: https://doi.org/10.7554/eLife.32319.012

nucleus anymore while the neighboring, more distant GFP-positive cell bodies contained elongated olMG nuclei (*Figure 3B–B''''*, see also *Figure 3—video 1*).

After incubation in BrdU for 3 days and fixation at 10 dpi, we observed similar results (*Figure 3C–3C''*). Here, we used immunohistochemistry to detect the olMG cell bodies via a GS-staining (*Figure 3C*). BrdU-positive cells in the ONL mark the site of the injury (*Figure 3C''*). In the region directly underneath the site of injury, the majority of olMG nuclei, which had been labeled by *rx2*::H2B-eGFP, were absent from the INL (*Figure 3C''*). GS-positive cell bodies remained spanning the entire apico-basal height, but without the apparent presence of nuclei. In contrast, unaffected GS-positive olMG cells located on either side of the wound site still contained their nuclei, as easily detected by the large size of the soma. This data shows that the cell bodies of injury-activated olMG cells are still intact despite the migration of their nuclei into the ONL.

## olMG cells divide in the INL with an apico-basal distribution

Since the injury response of olMG cells apparently does not involve self-renewal of olMG cells we wondered about the position and orientation of the cell division plane, a factor which has been associated with cell fate in various systems.

We first addressed the apico-basal position of dividing olMG cells by PH3 immunohistochemistry after injury. We detected PH3-positive cells only in the INL (*Figure 4A–4A''*). Some dividing cells were located more apically (*Figure 4A–4A''*), while others were located more basally (*Figure 4—figure supplement 1A–B*). This is in contrast to findings in zebrafish where, in a light injury paradigm, PH3-positive drMG cells can be found in the ONL 2 days after injury (*Nagashima et al., 2013*).

To address the cleavage plane of dividing olMG cells, we employed in vivo imaging of *rx2*::H2B-eGFP fish, which permits visualizing the separation of chromatin and thus gives a measurement of the orientation of division. The first injury-triggered olMG divisions were observed at 44 hpi (*Figure 4B–4B'''*, see also *Figure 4—video 1*). They occurred in the INL, both in the center and close to the ONL (*Figure 4—figure supplement 1C–C'''*). The mode of division was preferentially apico-basal (5 out of 6 divisions in 5 out of 6 animals), while only a single horizontal division was observed (1 out of 6 divisions in 1 out of 6 animals). In contrast, drMG cells are reported to predominantly divide with a horizontal division plane (*Lahne et al., 2015*). These results show that injury-induced olMG cell divisions occur in different positions in the INL and have a strong preference to occur apico-basally.

## olMG cells are lineage-restricted

In zebrafish, drMG cells are able to regenerate all neuronal cell types and self-renew after injury (*Nagashima et al., 2013*; *Powell et al., 2016*). We followed a BrdU-based lineage-tracing approach successfully applied in zebrafish (*Fausett and Goldman, 2006*; *Powell et al., 2016*) to address the potency of olMG cells. Transgenic *rx2*::H2B-eGFP fish retinae were injured either by two-photon laser ablation of PRCs or RGCs specifically or using a needle ablating all cell types. The injured fish were incubated in BrdU for 3 days to label proliferating cells. This allows to efficiently detect all injury-triggered S-phase entry of olMG cells (*Figure 5—figure supplement 1A–D*). For lineaging, fish were grown until 14 dpi to allow a regeneration response and subsequently analyzed for BrdU-positive cells in the different retinal layers (*Figure 5A*). PRC injuries led to the detection of 97% of all BrdU-positive cells in the ONL, mostly in the rod nuclear layer, indicative for PRC fate (*Figure 5B and E*). No BrdU-positive cells could be detected in the INL. Additionally, we found that the INL below the injury site was devoid of olMG cell nuclei, both consistently arguing for the absence of injury-triggered olMG self-renewal. Strikingly, RGC injuries did not trigger BrdU-uptake in olMG cells or any other differentiated cell type (data not shown). Needle injuries affecting all retinal cell types triggered the same response as the specific lesions in the PRC layer. 97% of all BrdU-positive cells were present in the ONL, and only a single BrdU-positive olMG cell was found in 1 of 10 fish

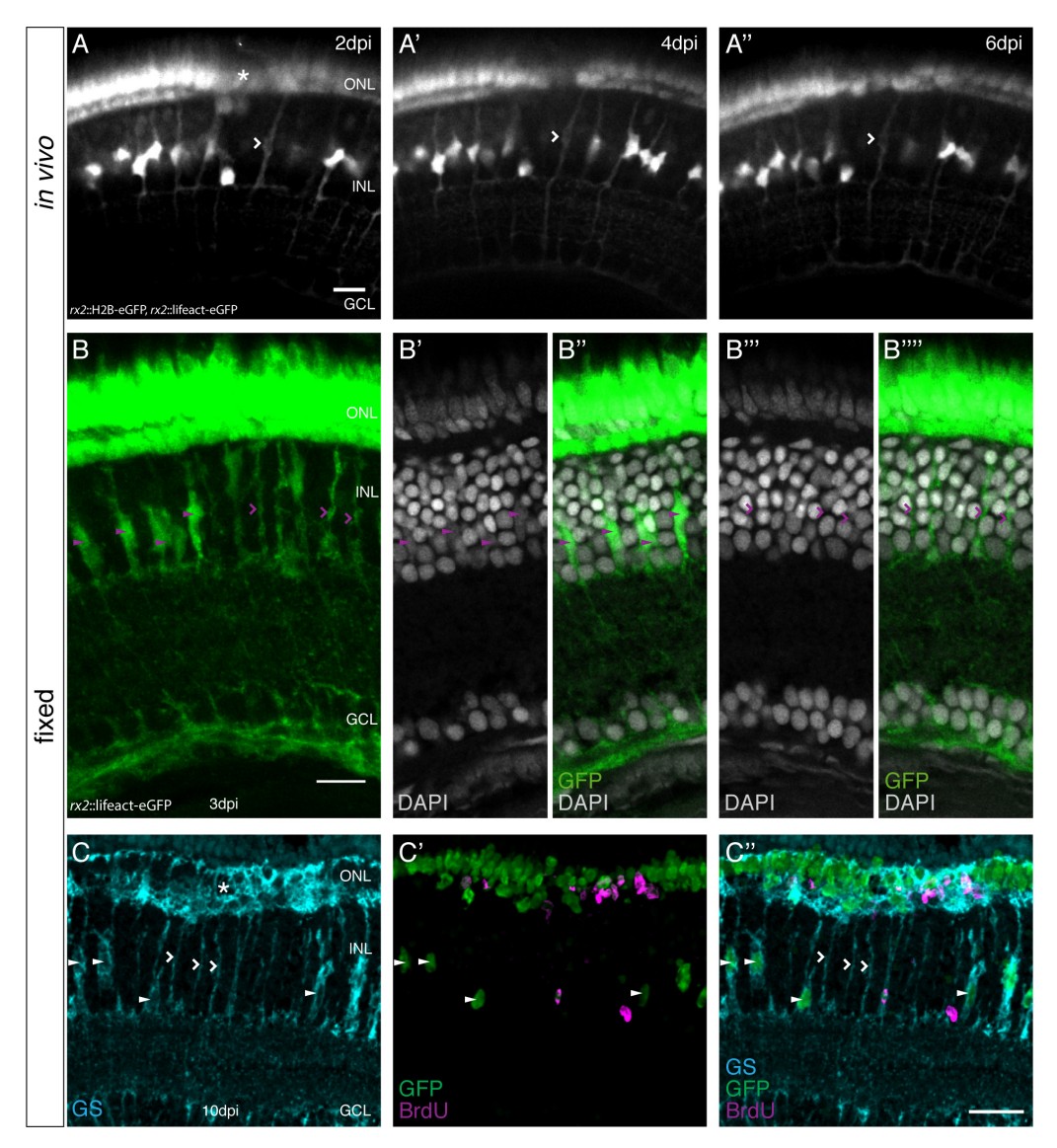

**Figure 3.** olMG nuclei are depleted after PRC injuries without cell body loss. (A–A'') In vivo imaging of a hatchling *rx2*::H2B-eGFP, *rx2*::lifeact-eGFP medaka retina which was injured in the ONL (asterisk) and imaged every second day after injury. Close to the injury site an olMG cell body without a nucleus can be detected at 2 dpi (A, empty arrowhead). The empty process remains until 6 dpi (A'') (n = 3 fish, data obtained from three independent experiments). Scale bar is 10 µm. (B–B'''') Maximum projection (B) and single planes of a cryosection of the injured hatchling medaka retina of the transgenic line *rx2*::lifeact-eGFP. The fish were injured, incubated in BrdU for 3 days and fixed at 3 dpi. Both GFP-positive cell bodies (green) which contain (arrowheads) and do not contain (empty arrowheads) a nucleus anymore are present. (n = 6 fish, data obtained from two independent experiments). Scale bar is 10 µm. (C–C'') Maximum projection of a cryosection of the injured hatchling medaka retina of the transgenic line *rx2*::H2B-eGFP. The fish were injured in the ONL (asterisk), incubated in BrdU for 3 days and fixed at 10 dpi. Many GFP-positive nuclei (green) are located in the ONL, some co-localizing with BrdU (magenta). In the INL few GFP-positive nuclei are present. Many GS-positive (cyan) olMG cell bodies below the injury site do not contain a GFP-positive nucleus (empty arrowheads). Next to the empty cell bodies GFP-positive nuclei can be detected within GS-positive cell bodies (arrowheads) (n = 4 fish, data obtained from two independent experiments). Scale bar is 20 µm.

DOI: https://doi.org/10.7554/eLife.32319.013

The following video is available for figure 3:

**Figure 3—video 1.** *rx2*::lifeact-eGFP retina at 3dpi

DOI: https://doi.org/10.7554/eLife.32319.014

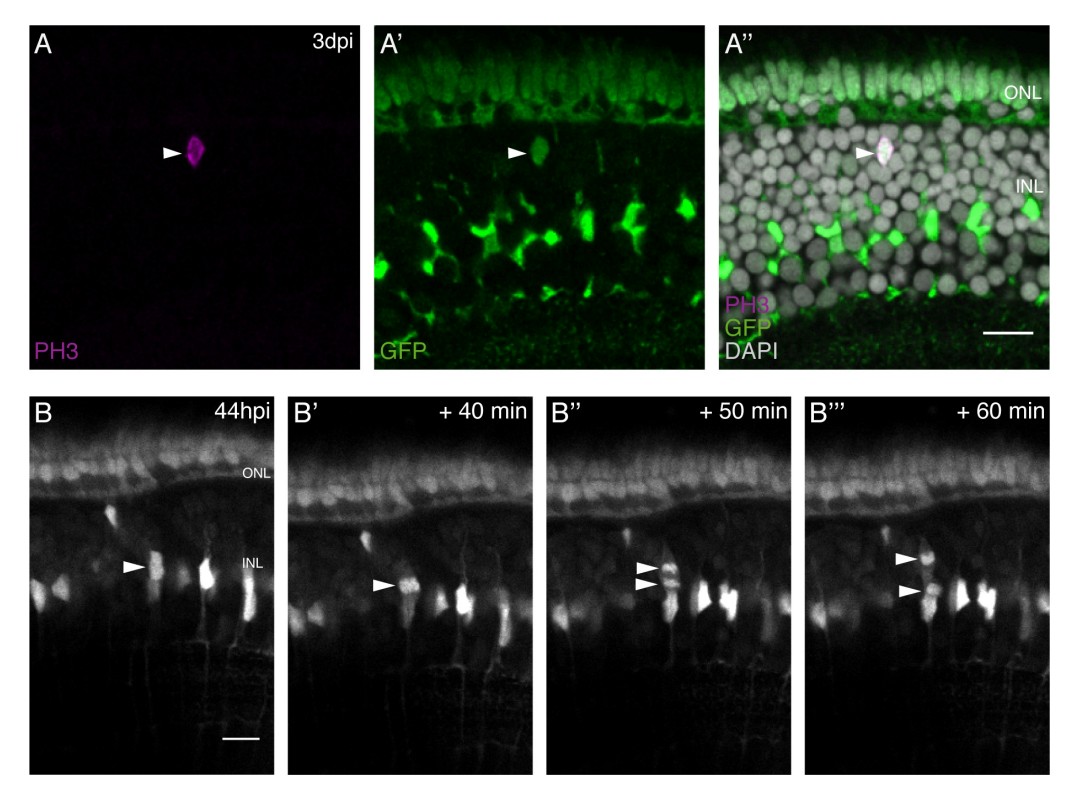

**Figure 4.** olMG cells divide in the INL with an apico-basal spindle orientation. (**A–A''**) Cryosection of an injured hatchling medaka retina of the transgenic line *rx2*::H2B-eGFP. PH3 stainings (magenta) on hatchling medaka retinae 3 days post PRC injury show mitotic olMG cells present in the INL (arrowhead), co-localizing with the *rx2* nuclear reporter expression (green) (n = 4 fish, data obtained from three independent experiments). (**B–B'''**) In vivo imaging of hatchling *rx2*::H2B-eGFP medaka retinae which were injured in the ONL and imaged starting at 44 hpi. OlMG nuclei which start to condense their chromatin can be detected in the INL (arrowheads). The divisions occur in an apico-basal manner (n = 6 fish, data obtained from six independent experiments, 5 out of 6 imaged divisions were apico-basal). Scale bars are 10 μm.

DOI: https://doi.org/10.7554/eLife.32319.015

The following video and figure supplement are available for figure 4:

**Figure supplement 1.** olMG cells divide in various positions in the the INL with an apico-basal spindle orientation.

DOI: https://doi.org/10.7554/eLife.32319.016

**Figure 4—video 1.** In vivo imaging of an olMG division after PRC injury

DOI: https://doi.org/10.7554/eLife.32319.017

(*Figure 5C and E*). Also later application of BrdU after injury (4 to 7 dpi) did not result in BrdU-positive olMG cells (*Figure 5—figure supplement 2A–C*). Importantly, BrdU-positive nuclei were not positive for GS, indicating that they were not olMG cells anymore (*Figure 5D*), but were positive for Recoverin, a PRC marker (*Figure 5E*). These results demonstrate that olMG cells do not self-renew and rather function as mono-potent repair system restricted to the generation of PRCs, most of which belong to the rod lineage.

### *Sox2* expression is not maintained in proliferating olMG cells after injury

The previous results show that olMG cells re-enter the cell cycle after injuries introduced by needle to the complete retina or by two-photon ablation to the PRC layer. They regenerate PRCs but do not undergo self-renewal. This suggests that olMG cells lack intrinsic factors that trigger self-renewal and multipotency upon injury. One transcription factor which is well known for its involvement in the self-renewal of stem cells – particularly neural stem cells – is Sox2 (*Sarkar and Hochedlinger, 2013*). It has been shown that cells expressing *sox2* are capable of both self-renewal and the production of a range of differentiated neuronal cell types (*Sarkar and Hochedlinger, 2013*). Data from zebrafish

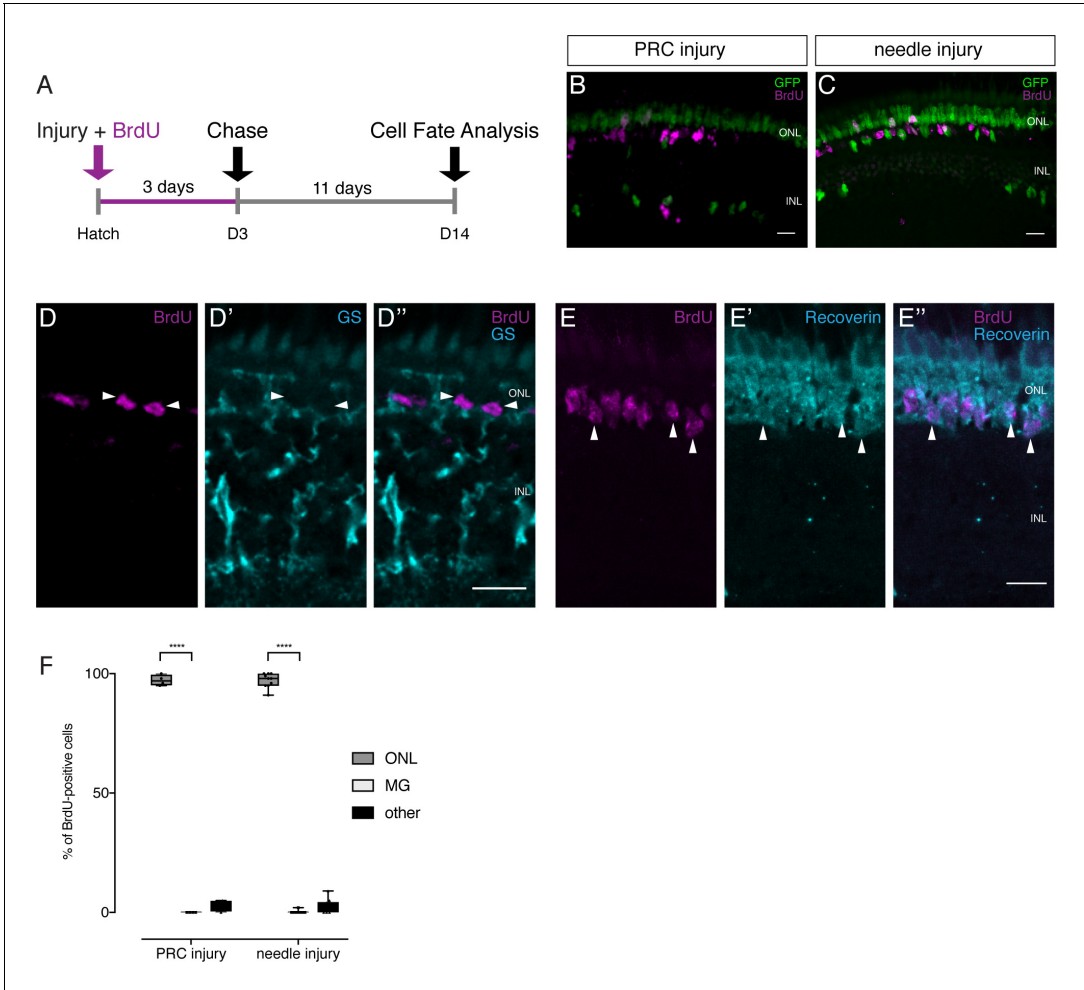

**Figure 5.** Lineage tracing after injuries reveals the preferential regeneration of PRCs. (**A**) Scheme outlining the experimental procedure. Hatchling medaka were injured in the retina with a two-photon laser ablating either PRCs or RGCs or with a needle ablating all cell types. The fish were incubated in BrdU for 3 days and analyzed at 14 dpi. (**B–C**) PRC injuries result in BrdU-positive cells in the ONL, mostly in the rod layer. No BrdU-positive olMG cells are present and fewer GFP-positive olMG cells are found in the INL (n = 4 fish, data obtained from two independent experiments). Needle injuries result in BrdU-positive cells in the ONL, mostly in the rod layer. Except for 1 BrdU-positive olMG cell in one fish, no BrdU-positive olMG cells are detected. GFP-positive olMG nuclei are largely depleted from the INL (n = 10 fish, data obtained from three independent experiments). (**D–D''**) After needle injuries BrdU-positive cells (magenta) in the ONL are not co-labeled with GS (cyan), indicating that they are not olMG cells (n = 8 fish, data obtained from two independent experiments). (**E–E''**) After needle injuries BrdU-positive cells (magenta) in the ONL are co-labeled with Recoverin (cyan), indicating that they are PRCs (n = 5 fish, data obtained from one experiment). Scale bars are 10 μm. (**F**) Quantification of the location of BrdU-positive cells reveals that in all injury types BrdU-positive cells are predominantly located in the ONL (PRC injury: 54 cells in four retinae, needle injury: 550 cells in 10 retinae). ****p<0.0001. Box plots: median, 25th and 75th percentiles; whiskers show maximum and minimum data points.

DOI: https://doi.org/10.7554/eLife.32319.018

The following figure supplements are available for figure 5:

**Figure supplement 1.** PRC and needle injuries trigger proliferation of olMG cells.
DOI: https://doi.org/10.7554/eLife.32319.019

**Figure supplement 2.** Late BrdU application after injury labels the same cell population as early BrdU application.
DOI: https://doi.org/10.7554/eLife.32319.020

have shown that a ubiquitous gain of *sox2* expression triggers a proliferative response of drMGs in the absence of injury (*Gorsuch et al., 2017*).

To investigate the expression of *sox2* in MG cells, we performed immunohistochemistry on uninjured retinae in medaka and zebrafish. In the medaka retina, Sox2 protein is detected in amacrine cells (ACs) and olMG cells in the central retina (*Figure 6A–6A'''*). In zebrafish, the pattern is similar: Sox2 protein is present in ACs and drMG cells in the central retina (*Figure 6B–6B'''*). This data is

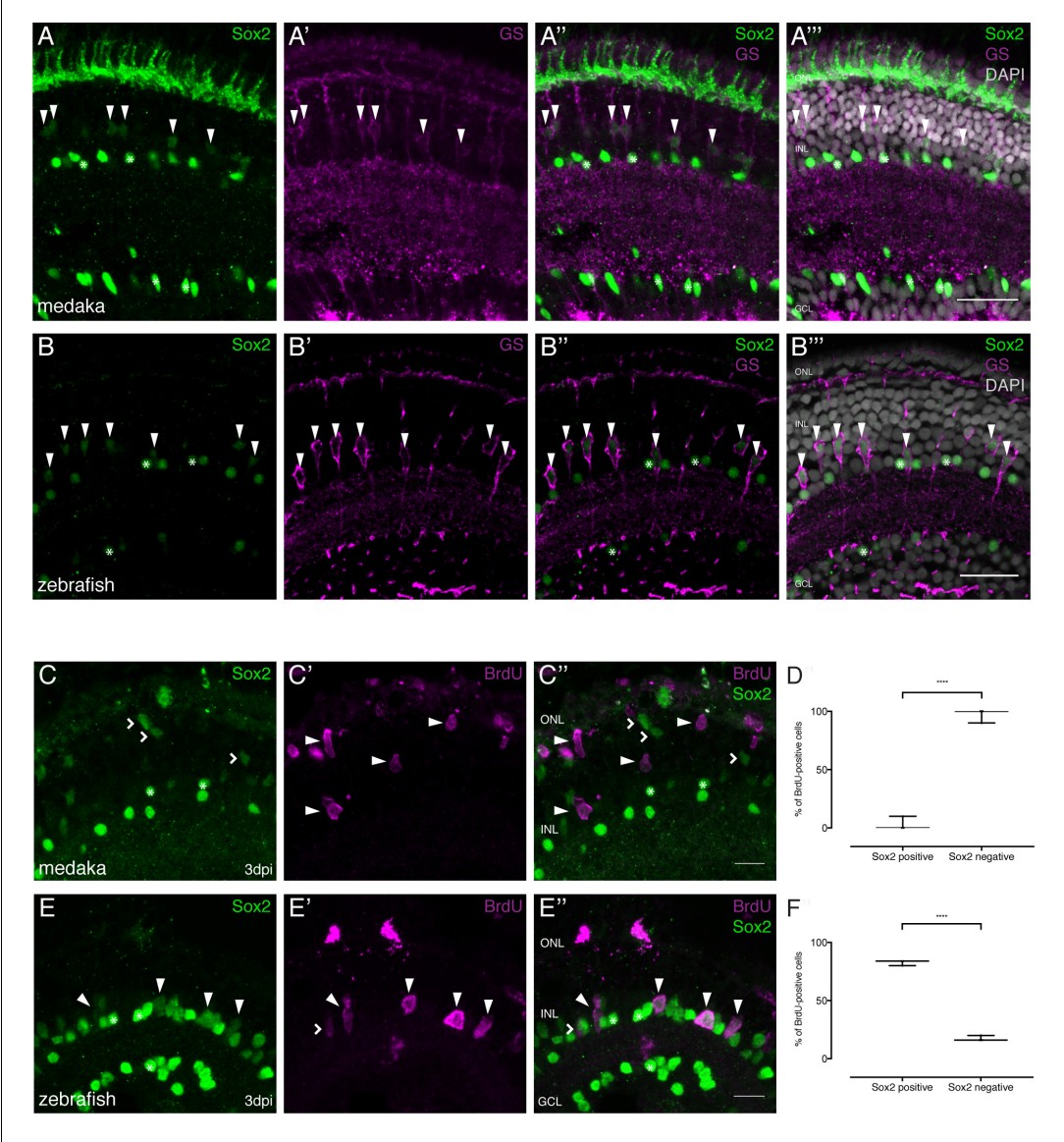

**Figure 6.** Sox2 is present in MG cells of the hatchling medaka and zebrafish retina but not maintained after injury in medaka. (A–A''') Cryosection of an uninjured hatchling medaka retina. Sox2 (green) labeled cells with round nuclei are present in the INL and the GCL. Sox2-labeled cells with round nuclei are ACs present in the INL and the GCL (asterisks). Sox2-positive cells with elongated nuclei are present in the INL (arrowheads). Co-labeling with GS (magenta) proves that cells with elongated nuclei are olMG cells. Additional staining, which is likely unspecific staining since *sox2* mRNA cannot be detected there (*Reinhardt et al., 2015*), can be detected in the ONL. (B–B''') Cryosection of an uninjured zebrafish retina at 9 dpf. Sox2 (green)-positive cells with round nuclei are present in the INL and the GCL. Sox2-labeled cells with round nuclei are ACs present in the INL and the GCL (asterisks). Sox2-positive cells with elongated nuclei are present in the INL (arrowheads). Co-labeling with GS (magenta) proves that cells with elongated nuclei are drMG cells. Scale bars are 20 µm. (C–C'') Cryosection of an injured hatchling medaka retina at 3 dpi. BrdU (magenta, arrowheads) labeled cells are not co-labeled with Sox2 (green, arrowheads). Sox2-positive cells with elongated nuclei, indicating non-proliferative olMG cells, are found in the INL (open arrowheads). Sox2-positive cells with round nuclei are ACs present in the INL and the GCL (asterisks) (n = 3 fish, data obtained from two independent experiments). (D) Quantification of the amount of Sox2-positive and negative proliferating cells of BrdU-positive cells at 3 dpi in medaka (74 cells in 3 retinae). ****p<0.0001. Box plots: median, 25th and 75th percentiles; whiskers show maximum and minimum data points. (E–E'') Cryosection of an injured zebrafish retina at 3 dpi. BrdU-(magenta) and Sox2-(green) double positive cells can be detected in the INL (arrowheads). BrdU-positive Sox2-negative cells can rarely be detected (open arrowhead). Sox2-labeled cells with round nuclei are ACs present in the INL and the GCL (asterisks) (n = 3 fish, data obtained from two independent experiments). Scale bars are 10 µm. (F) Quantification of the amount of Sox-positive and negative proliferating cells of BrdU-positive cells at 3 dpi in zebrafish (68 cells in 3 retinae). ****p<0.0001. Box plots: median, 25th and 75th percentiles; whiskers show maximum and minimum data points.

DOI: https://doi.org/10.7554/eLife.32319.021

consistent with data from other vertebrates including human, whose MG cells also maintain *sox2* expression (*Gallina et al., 2014*).

To investigate the expression of *sox2* in the olMG and drMG cells responding to injury by proliferation, we performed needle injuries, incubated the fish in BrdU and fixed them at 3 dpi. We could detect BrdU-positive MG cells both in medaka and zebrafish. The vast majority of proliferating olMG cells did not express *sox2* anymore at 3 dpi (*Figure 6C–6D*, 6% of all BrdU-positive cells were Sox2-positive). We saw a similar scenario in response to either PRC or RGC injury, where 9% and 10% respectively of all BrdU-positive cells were Sox2-positive. Conversely, in zebrafish, *sox2* expression was still detected after 3 days in drMG cells that proliferated in response to needle injury (*Figure 6E–6F*, 84% of all BrdU-positive cells were Sox2-positive). These findings strongly argue that the downregulation of *sox2* expression in proliferating olMG cells restricts their regenerative properties.

## Sustained Sox2 expression restores olMG-driven regeneration

The results presented above indicate that after injury, olMG cells and olMG-derived progenitors do not maintain the expression of *sox2*, in contrast to the situation in zebrafish. We hypothesize that the prolonged *sox2* expression facilitates drMG cells to undergo self-renewal and to generate neurogenic clusters and ultimately all cell types necessary to regenerate a functional retina. To test this hypothesis, we chose the inducible LexPR transactivation system (*Emelyanov and Parinov, 2008*) targeted to olMG cells (*rx2*::LexPR OP::*sox2*, OP::H2B-eGFP) to sustain *sox2* expression. In mifepristone-treated retinae, we detected increased levels of Sox2 protein in induced olMG cells (*Figure 7A–B''*). To address the proliferative behavior of Sox2-sustaining olMG cells in response to injury, we treated fish with mifepristone and BrdU for 2 days, performed a needle injury, maintained the fish in mifepristone and BrdU until 3 dpi and analyzed immediately (*Figure 7C*). We observed increased formation of proliferating clusters as well as the distribution of BrdU-positive cells in all layers of the retina after needle injury (4 out of 6 fish) (*Figure 7D–E'*). To address the long-term potential of Sox2-induced olMG cells, we ablated all retinal cell types by needle injury and performed BrdU-mediated lineage tracing as described above. We induced *sox2* expression for 2 days and provided BrdU in parallel, performed a needle injury and maintained the expression of *sox2* until 3 dpi. After a chase until 14 dpi, the retinae and regenerated cell types were analyzed (*Figure 8A and B*). In needle-injured wild-type fish which were also treated with mifepristone as well as in transgenic fish (*rx2*::LexPR OP::*sox2*, OP::H2B-eGFP) which were not treated with mifepristone, olMG cells did not self-renew and gave predominantly rise to PRCs, mostly rod PRCs (*Figure 8F*). In contrast, olMG cells experiencing persistent expression of *sox2* showed self-renewal and differentiation into different cell types in the ONL and INL as indicated by BrdU lineage tracing. In particular, the olMG cells maintaining *sox2* expression after injury regenerated olMG cells (*Figure 8C–C'' and F*) and exhibited a significant increase in regenerated ACs and RGCs, which were positive for HuC/D (*Figure 7D–E'' and 8F*). Furthermore, a slight increase in cone PRCs and a decrease in rod PRCs was observed after *sox2* induction (*Figure 8C–F*). These data indicate that a targeted maintenance of *sox2* expression after injury is sufficient to induce self-renewal and increase potency in olMG cells in the medaka retina turning a mono-potent repair system into a regeneration system with increased potency.

## Discussion

Here, we have characterized a differential regenerative response between two teleost fish and used it as a framework to address the molecular determinants of regeneration during evolution. By using a combination of in vivo imaging, targeted cell type ablation and lineage tracing, we investigated the dynamics of the injury response in the medaka retina. We focused on MG cells, which play a prominent role in zebrafish retinal regeneration. While upon injury olMG cells re-enter the cell cycle, they fail to undergo self-renewal. Furthermore, olMG cells do not generate the neurogenic clusters which arise in zebrafish, nor do they produce all neuronal cell types in the retina. We traced this effect prominently to Sox2, the expression of which is maintained in proliferating drMG cells after injury, but not in olMG cells. We demonstrated that the sustained expression of *sox2* is sufficient to convert an olMG into a dr-like MG cell. The fact that this response is acquired cell-autonomously and in the context of a non-regenerative retina can be relevant for putative translational approaches.

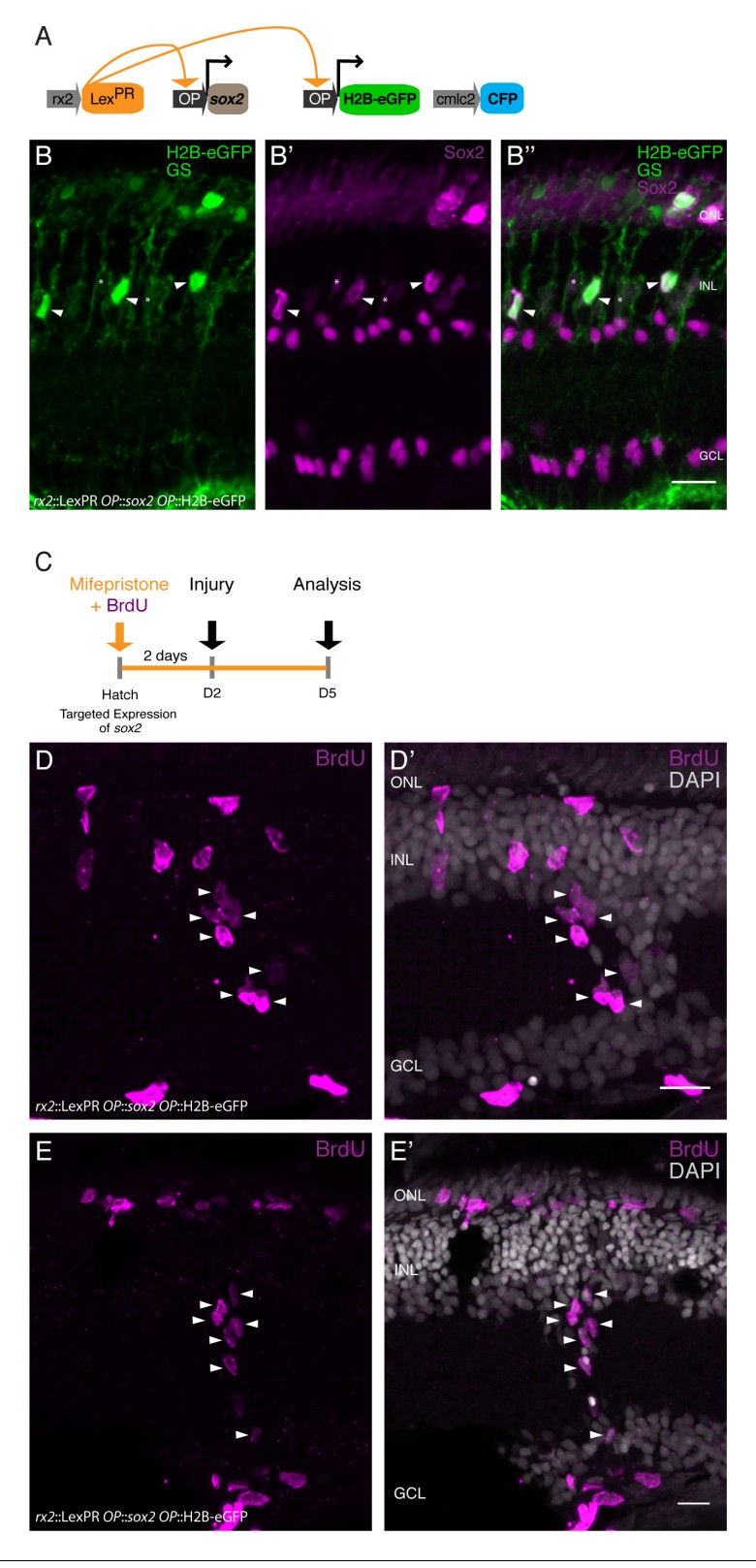

**Figure 7.** Expression of *sox2* via the LexPR system increases Sox2 protein levels in olMG cells and triggers proliferating cluster formation after injury in medaka. (**A**) Genetic construct used for *sox2* induction. (**B–B''**) Cryosection of a retina of a mifepristone-induced *rx2*::LexPR OP::*sox2*, OP::H2B-eGFP transgenic fish at 5 days after induction. Nuclear-localized GFP (green) labels positively induced cells, which contain an increased amount

*Figure 7 continued on next page*

*Figure 7 continued*

of Sox2 protein (magenta, arrowheads) in comparison to non-induced cells (asterisks) (n = 6 fish, data obtained from two independent experiments). (C) Induction scheme for *sox2* induction. (D–E') Cryosections of mifepristone-induced *rx2*::LexPR OP::*sox2*, OP::H2B-eGFP transgenic fish at 3 days after needle injury. BrdU-positive (magenta) cells can be detected in all retinal layers and some BrdU-positive clusters are present in the INL and between INL and GCL (n = 4 fish, data obtained from one experiment). Scale bars are 10 µm.

DOI: https://doi.org/10.7554/eLife.32319.022

Since olMG cells do not self-renew after injuries and only have the capacity to regenerate PRCs, olMG cells are not true multipotent retinal stem cells. Instead, olMG cells should be considered lineage-restricted progenitors. They re-enter the cell cycle between 1 and 2 dpi, similar to the re-entry observed in zebrafish. This indicates that the signals that are essential for cell cycle re-entry are present in medaka and are activated in a window of time similar to that in zebrafish. After retinal injures olMG nuclei migrate into the PRC layer; the cell bodies of nucleus-depleted olMG cells are maintained in the retina. This could be important since MG cell bodies play a crucial role in mechanical stability of the retina (*MacDonald et al., 2015*) as well as light guiding through the retina (*Franze et al., 2007*). After retinal injures, olMG cell bodies were maintained in the absence of a nucleus in the INL reflecting the necessity to preserve this structural and optical element.

In the uninjured retina, olMG cells express *sox2*, as is the case for many other vertebrates, including humans. However, *sox2* expression in olMG cells is downregulated in response to injury, in contrast to the injury response of drMG cells, which upregulate *sox2* (*Gorsuch et al., 2017*). We speculate that this upregulation is due to epigenetic modifications of the *sox2* locus. A recent study in the mouse retina showed that the expression of *oct4* is upregulated shortly after injury and then downregulated at 24 hpi (*Reyes-Aguirre and Lamas, 2016*). This correlates with a decrease in the expression of DNA methyltransferase 3b and its subsequent upregulation at 24 hpi, triggering a decrease in methylation and subsequent re-methylation of *oct4*. Furthermore, a recent study on zebrafish regeneration discovered the existence of so-called tissue regeneration enhancer elements (TREEs) (*Kang et al., 2016*). One TREE was associated with *leptin b*, which is expressed in response to injuries of the fin and heart. This TREE acquires open chromatin marks after injury, can be divided into tissue-specific modules and can drive injury-dependent expression in mouse tissue. This raises the possibility that the *sox2* locus in olMG cells experiences epigenetic modifications after injury which differ from modifications in zebrafish. The fact that *sox2* expression is detected in all vertebrate MG cells analyzed to date in the absence of injury raises the question whether a decrease in *sox2* expression after injury might be a common feature of non-regenerative species like chicken, mouse and even humans. Data from a conditional *sox2* knockout in mouse shows that Sox2 is necessary for maintenance of MG morphology and quiescence (*Surzenko et al., 2013*). While its expression is maintained in response to the injection of growth factors after retinal damage (*Karl et al., 2008*) its regulation in response to injury alone has not been described. Data obtained in cultures of human MG cells (*Bhatia et al., 2011*) provide additional important insights. Strikingly similar to medaka, silencing the expression of *sox2* caused MG cells to lose stem and progenitor cell markers and adopt a neural phenotype (*Bhatia et al., 2011*). These findings align well with the results from medaka presented here und suggest that olMG cells and their behavior as progenitor cells can serve as a model for mammalian and in particular human MG cells.

The results shown here may provoke an evolutionary question: is retinal regeneration an ancestral or derived feature within the infraclass of teleosts? The question might be resolved by investigations of this capacity in other fish species more closely related to medaka, such as *Xiphophorus maculatus*, whose last common ancestor with medaka lived around 120 million years ago (*Schartl et al., 2013*). Additionally, species like the spotted gar, whose lineage diverged from teleosts before teleost genome duplication (*Braasch et al., 2016*), might provide insights about the ancestral mode of retinal regeneration. Recently, the retinal architecture of the spotted gar has been analyzed (*Sukeena et al., 2016*). There, proliferative cells have been detected in the central retina likely representing proliferating MG cells, which generate rod PRCs during homeostasis as seen in zebrafish, suggesting that regeneration is indeed an ancestral feature in the ray-finned fish lineage.

Additionally, one wonders whether the ability of MG cells to regenerate injured retinal cells is directly related to their involvement in rod genesis during post-embryonic growth and conversely

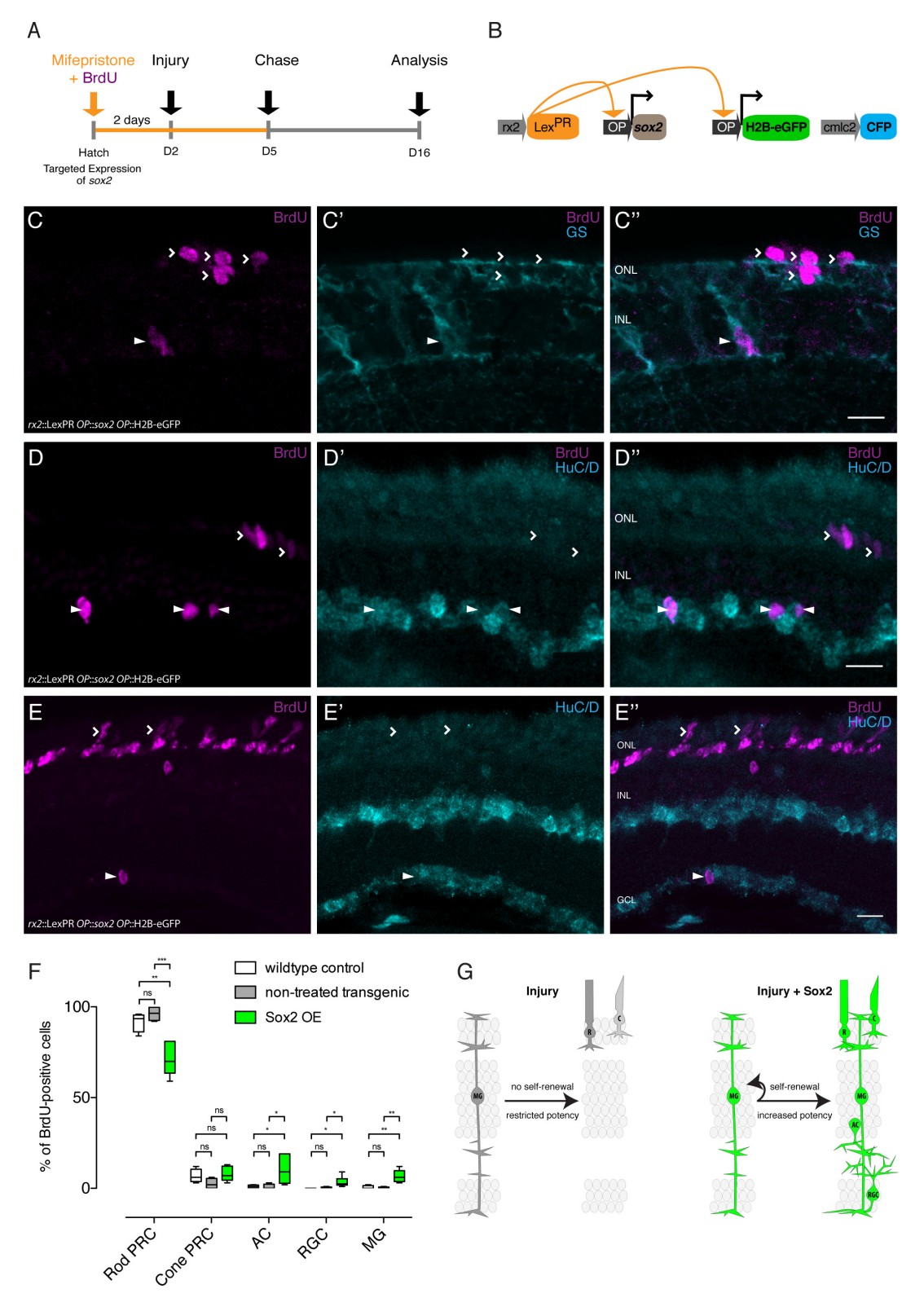

**Figure 8.** Sox2 induces a regeneration response in olMG cells. (**A–B**) Induction scheme and construct (**B**) used for *sox2* induction. (**C–C''**) Cryosection of a *sox2*-induced hatchling medaka retina. BrdU-positive (magenta) olMG cells, which are labeled by GS (cyan) can be detected in the INL (arrowhead). Additional BrdU-positive cells are located in the ONL, in the location of both rods and cones (open arrowheads). (**D–E''**) Cryosections of *sox2*-induced hatchling medaka retinae. BrdU-positive (magenta) ACs, which are labeled by HuC/D (cyan) can be detected in the INL (**D-D''**, arrowheads) and GCL (**E**-

*Figure 8 continued on next page*

*Figure 8 continued*

E'', arrowheads). Additional BrdU-positive cells are located in the ONL, in the location of both rods and cones (open arrowheads) (n = 7 sox2 OE fish and n = 8 control fish, data obtained from two independent experiments each). Scale bars are 10 μm. (F) Quantification of the location of BrdU-positive cells in *sox2*-induced fish (607 cells in 14 retinae) versus wild-type control fish treated with mifepristone (341 cells in eight retinae) and transgenic *rx2*::LexPR OP::*sox2*, OP::H2B-eGFP fish not treated with mifepristone (218 cells in four retinae) reveals an increase in BrdU-positive olMG cells, ACs and RGCs as well as a decrease in rod PRCs in *sox2*-induced fish. Wild-type control vs Sox2 OE: Rod PRC **p=0.0031, Cone PRC ns p=0.678, AC *p=0.0434, RGC *p=0.0445, MG **p=0.0083. Non-treated transgenic vs Sox2 OE: Rod PRC ***p=0.0004, Cone PRC ns p=0.528, AC *p=0.0445, RGC *p=0.0445, MG **p=0.0061. Box plots: median, 25th and 75th percentiles; whiskers show maximum and minimum data points. (G) olMG cells respond to injuries by proliferation without self-renewal and restriction toward PRC fate. Targeted expression of *sox2* induces self-renewal and increased potency of olMG cells.

DOI: https://doi.org/10.7554/eLife.32319.023

whether the lack of regenerative abilities of MG cells is a result of the lack of rod genesis? The differences in rod layer increase in zebrafish and medaka as well as the differences in rod genesis by MG cells might be due to the natural habitats and photic environment of the fish. While larval zebrafish live near the water surface, adults live in deeper waters where rods become more important for visual function (*Lenkowski and Raymond, 2014*). Medaka on the other hand are surface fish their entire life, since they live in shallow waters like rice paddies (*Kirchmaier et al., 2015*) which decreases the need for a massive increase in rods.

With a potential translational perspective, regenerating and non-regenerating systems can now be systematically compared to delineate the underlying factors and mechanisms.

To date, our cumulative results show that the regenerative potential of olMG cells in the context of homeostasis and injury in medaka resemble that of mammals and birds more than zebrafish. We propose that this provides an added value to medaka as a model species for regeneration studies that bridge the differences between zebrafish and mammals. Studies of heart regeneration that have compared zebrafish and medaka lend additional support to this statement (*Ito et al., 2014*; *Lai et al., 2017*). As reprogrammable multipotent retinal stem cells, MG cells harbor a great potential for treating degenerative retinal diseases. Our work indicates that the addition of a single reprogramming factor facilitates a regeneration-like response mediated by olMG cells. Their multiple resemblances of features of mammalian and human MG cells position them as an ideal model for the development of new treatments preventing the degeneration and initiating the regeneration of the retina.

## Materials and methods

**Key resources table**

| Reagent type (species) or resource | Designation | Source or reference | Identifiers | Additional information |
|---|---|---|---|---|
| Strain (*Oryzias latipes*) | Cab | other | | medaka Southern wild type population |
| Strain (*O. latipes*) | *rx2*::H2B-eGFP | this paper | | |
| Strain (*O. latipes*) | *rx2*::lifeact-eGFP | this paper | | |
| Strain (*O. latipes*) | *rx2*::H2B-eGFP QuiH | this paper | | |
| Strain (*O. latipes*) | *rx2*::LexPR *OP*::*sox2* *OP*::H2B-eGFP *cmlc2*::CFP | this paper | | |
| Strain (*O. latipes*) | GaudíRSG | PMID: 25142461 | | |
| Strain (*O. latipes*) | *rx2*::CreERT2 | PMID: 25908840 | | |

*Continued on next page*

Continued

| Reagent type (species) or resource | Designation | Source or reference | Identifiers | Additional information |
|---|---|---|---|---|
| Strain (*Danio rerio*) | AB | other | | Wildtype zebrafish strain |
| Strain (*D. rerio*) | Albino | other | | |
| Antibody | anti-BrdU (rat) | Bio-Rad/AbD Serotec (Germany) | BU1/75, RRID: AB_609566 | 1: 200 |
| Antibody | anti-eGFP (chicken/IgY, polyclonal) | Thermo Fisher (Waltham, Massachusetts, USA) | A10262, RRID: AB_2534023 | 1: 500 |
| Antibody | anti-HuC/D (mouse, monoclonal) | Thermo Fisher | A21271, RRID: AB_221448 | 1: 200 |
| Antibody | anti-GS (mouse, monoclonal, clone GS-6) | Milipore (Burlington, Massachusetts, USA) | MAB302, RRID: AB_2110656 | 1: 500 |
| Antibody | anti-pH3 (Ser10) (rabbit, polyclonal) | Millipore | 06–570, RRID: AB_310177 | 1: 500 |
| Antibody | anti-Recoverin (rabbit, polyclonal) | Millipore | AB5585, RRID: AB_2253622 | 1: 200 |
| Antibody | anti-Sox2 (rabbit, polyclonal) | Genetex (Irvine, California, USA) | GTX101506, RRID: AB_2037810 | 1: 100 |
| Antibody | anti-Zpr-1 (mouse, Imonoclonal) | ZIRC (Eugene, Oregon, USA) | RRID: AB_10013803 | 1: 200 |
| Antibody | anti-chicken Alexa Fluor 488 (donkey) | Jackson (West Grove, Pennsylvania, USA) | 703-545-155, RRID: AB_2340375 | 1: 750 |
| Antibody | anti-mouse Alexa Fluor 546 (goat) | Thermo Fisher | A-11030, RRID: AB_2534089 | 1: 750 |
| Antibody | anti-mouse Cy5 (donkey) | Jackson | 715-175-151, RRID: AB_2340820 | 1: 750 |
| Antibody | anti-rabbit DyLight 549 (goat) | Jackson | 112-505-144 | 1: 750 |
| Antibody | anti-rabbit Alexa Fluor 647 (goat) | Thermo Fisher | A-21245, RRID: AB_2535813 | 1: 750 |
| Antibody | anti-rat DyLight 549 (goat) | Jackson | 112-505-143 | 1: 750 |
| Antibody | anti-rat Alexa Fluor 633 (goat) | Thermo Fisher | A-21094, RRID: AB_2535749 | 1: 750 |
| Recombinant DNA reagent | *rx2*::H2B-eGFP (plasmid) | this paper | | Vector with I-SceI meganuclease sites |
| Recombinant DNA reagent | *rx2*::lifeact-eGFP (plasmid) | this paper | | Vector with I-SceI meganuclease sites |
| Recombinant DNA reagent | *rx2*::LexPR OP::*sox2* OP (plasmid) | this paper | | Vector with I-SceI meganuclease sites |
| Recombinant DNA reagent | OP::H2B-eGFP cmlc2::CFP (plasmid) | this paper | | Vector with I-SceI meganuclease sites |
| Sequence-based reagent | PRC primer for medaka *sox2* | | | fwd with BamHI site: TAATGGATCCATG TATAACATGATG GAGACTGAAC, rev with NotI site: TAATGCGGCCGCT TACATGTGTGTTAACGGCAGCGTGC |

*Continued*

| Reagent type (species) or resource | Designation | Source or reference | Identifiers | Additional information |
|---|---|---|---|---|
| Chemical compund, drug | 5-Bromo-2′-deoxyuridine (BrdU) | Sigma Aldrich (St. Louis, Missouri, USA) | B5002 | |
| Chemical compund, drug | Mifepristone | Cayman (Ann Arbor, Michigan, USA) | 84371-65-3 | |
| Chemical compund, drug | 1-phenyl-2-thiourea (PTU) | Sigma Aldrich | P7629 | |
| Chemical compund, drug | Tamoxifen | Sigma Aldrich | T5648 | |
| Chemical compund, drug | Tricaine | Sigma Aldrich | A5040 | |
| Other | DAPI | Roth (Germany) | 28718-90-3 | 1:500 dilution in 1xPTW of 5 mg/ml stock |

## Animals and transgenic lines

Medaka (*Oryzias latipes*) and zebrafish (*Danio rerio*) used in this study were kept as closed stocks in accordance to Tierschutzgesetz 111, Abs. 1, Nr. 1 and with European Union animal welfare guidelines. Fish were maintained in a constant recirculating system at 28°C on a 14 hr light/10 hr dark cycle (Tierschutzgesetz 111, Abs. 1, Nr. 1, Haltungserlaubnis AZ35–9185.64 and AZ35–9185.64/BH KIT). The following stocks and transgenic lines were used: wild-type Cabs, *rx2*::H2B-eGFP, *rx2*::lifeact-eGFP, *rx2*::H2B-eGFP QuiH, *rx2*::LexPR *OP*::sox2 *OP*::H2B-eGFP *cmlc2*::CFP, *rx2*::CreERT2, GaudíRSG (*Reinhardt et al., 2015*), AB zebrafish and Albino zebrafish. All transgenic lines were created by microinjection with Meganuclease (I-SceI) in medaka embryos at the one-cell stage, as previously described (*Thermes et al., 2002*).

## BrdU incorporation

For BrdU incorporation, fish were incubated in 2.5 mM BrdU diluted in 1x Embryo Rearing Medium (ERM) or 1x Zebrafish Medium for respective amounts of time.

## Induction of the LexPR system and induction of Cre/lox system

For induction of the LexPR system, fish were induced by bathing them in a 5 µM to 10 µM mifepristone solution in 1x ERM for respective times. For induction of the Cre/lox system, fish were treated with a 5 µM tamoxifen solution in 1x ERM over night.

## In vivo imaging and laser ablations

For in vivo imaging fish in a Cab background were kept in 5x 1-phenyl-2-thiourea (PTU) in 1x ERM from 1 dpf until imaging to block pigmentation. Fish in a QuiH background could be imaged without any treatment. Fish were anesthetized in 1x Tricaine diluted in 1x ERM and mounted in glass-bottomed Petri dishes (MatTek Corporation, Ashland, MA) in 1% Low Melting Agarose. The specimens were oriented lateral, facing down, so that the right eye was touching the cover-slip at the bottom of the dish. Imaging and laser ablations were performed on a Leica (Germany) SP5 equipped with a Spectra Physics (Santa Clara, California, USA) Mai Tai HP DeepSee Ti:Sapphire laser, tunable from 690 to 1040 nm and Leica Hybrid Detectors. A wound was introduced using the bleach point function or the region of interest function, together with the high-energy two-photon laser tuned to 880 nm. The wound size was defined between 40 and 60 µm diameter for medium-sized wounds. Wounds bigger than 60 µm diameter were defined as large wounds. *Rx2*::H2B-eGFP or *rx2*::lifeact-eGFP fish were used for the ablations. Since *rx2* is expressed during retinal development residual GFP could be visualized in rod PRCs as well as in RGCs when increasing the gain of the Hybrid Detectors. Follow-up imaging was performed using same laser at 880 nm and a 40x objective.

## Retinal needle injuries

Larvae (zebrafish 5 dpf, medaka 8 dpf) were anesthetized in 1x Tricaine in 1x ERM and placed on a wet tissue. Under microscopic visualization, the right retina was stabbed multiple times in the dorsal part with a glass needle (0.05 mm diameter). Left retinae were used as controls.

## Immunohistochemistry on cryosections

Fish were euthanized using Tricaine and fixed over night in 4% PFA, 1x PTW at 4°C. After fixation samples were washed with 1x PTW and cryoprotected in 30% sucrose in 1x PTW. To improve section quality, the samples were incubated in a half/half mixture of 30% sucrose and Tissue Freezing Medium (Leica) for at least 3 days. 16-µM-thick serial sections were obtained on a Leica cryostat. Sections were rehydrated in 1x PTW for 30 min at room temperature. Blocking was performed for 1–2 hr with 10% NGS (normal goat serum) in 1x PTW at room temperature. The respective primary antibodies were applied diluted in 1% NGS o/n at 4°C. The secondary antibody was applied in 1% NGS together with DAPI (1:500 dilution in 1xPTW of 5 mg/ml stock) for 2–3 hr at 37°C. Slides were mounted with 60% glycerol and kept at 4°C until imaging.

## Antibodies

| Primary antibody | Species | Concentration | Company |
| --- | --- | --- | --- |
| Anti-BrdU | rat | 1:200 | AbD Serotec, BU1/75 |
| Anti-eGFP | chicken | 1:500 | Thermo Fisher, A10262 |
| Anti-HuC/D | mouse | 1:200 | Thermo Fisher, A21271 |
| Anti-GS | mouse | 1:500 | Chemicon, MAB302 |
| Anti-pH3 (Ser10) | rabbit | 1:500 | Millipore, 06–570 |
| Anti-Recoverin | rabbit | 1:200 | Millipore, AB5585 |
| Anti-Sox2 | rabbit | 1:100 | Genetex, GTX101506 |
| Anti-Zpr-1 | mouse | 1:200 | ZIRC |

| Secondary antibody | Species | Concentration | Company |
| --- | --- | --- | --- |
| Anti-chicken Alexa Fluor 488 | donkey | 1:750 | Jackson, 703-485-155 |
| Anti-mouse Alexa 546 | goat | 1:750 | Thermo Fisher, A-11030 |
| Anti-mouse Cy5 | donkey | 1:750 | Jackson, 715-175-151 |
| Anti-rabbit DyLight549 | goat | 1:750 | Jackson, 112-505-144 |
| Anti-rabbit 647 | goat | 1:750 | Thermo Fisher, A-21245 |
| Anti-rat DyLight549 | goat | 1:750 | Jackson, 112-505-143 |
| Anti-rat Alexa633 | goat | 1:750 | Thermo Fisher, A-21094 |

## BrdU immunohistochemistry

BrdU antibody staining was performed with an antigen retrieval step. After all antibody stainings and DAPI staining, except for BrdU, were complete, a fixation for 30 min was performed with 4% PFA. Slides were incubated for 1 hr at 37°C in 2 N HCl solution, and pH was recovered by washing with a 40% Borax solution before incubation with the primary BrdU antibody.

## TUNEL staining

TUNEL stainings on cryosections were performed after all other antibody stainings were completed using the In Situ Cell Death Detection Kit TMR Red by Roche. Stainings were performed according to the manufacturers protocol with the following modifications. Washes were performed with 1x PTW instead of PBS.

## Immunohistochemistry imaging

All immunohistochemistry images were acquired by confocal microscopy at a Leica TCS SPE with either a 20x water objective or a 40x oil objective.

## Image processing and statistical analysis

Images were processed via Fiji image processing software. Statistical analysis and graphical representation of the data were performed using the Prism software package (GraphPad). Box plots show the median, 25th and 75th percentiles; whiskers show maximum and minimum data points. Unpaired t-tests were performed to determine the statistical significances. The p-value $p < 0.05$ was considered significant and p-values are given in the figure legends. Sample size (n) and number of independent experiments are mentioned in every figure legend. No statistical methods were used to predetermine sample sizes, but our sample sizes are similar to those generally used in the field. The experimental groups were allocated randomly, and no blinding was done during allocation.

## Acknowledgements

We thank the Wittbrodt department for constructive discussions on the project; L Centanin and A Gutierrez-Triana for valuable input on the project and the manuscript; N Aghaallaei, C Becker, F Caroti, A-K Heilig, I Krämer, S Lemke, C Lischik, T Tavhelidse and E Tsingos for critical reading of the manuscript; R Hodge for manuscript editing. We are grateful to A Saraceno, E Leist and M Majewski for fish husbandry. KL is a member of HBIGS, the Heidelberg Graduate School for Life Sciences and was supported by a LGFG Fellowship. This work was supported by the European Research Council (GA 294354-ManISteC to JW)

## Additional information

### Funding

| Funder | Grant reference number | Author |
|---|---|---|
| European Commission | Advanced Grant 294354 | Joachim Wittbrodt |

The funders had no role in study design, data collection and interpretation, or the decision to submit the work for publication.

### Author contributions

Katharina Lust, Conceptualization, Data curation, Formal analysis, Validation, Investigation, Visualization, Methodology, Writing—original draft, Writing—review and editing; Joachim Wittbrodt, Conceptualization, Funding acquisition, Writing—original draft, Project administration, Writing—review and editing

### Author ORCIDs

Katharina Lust (iD) http://orcid.org/0000-0002-2580-5492
Joachim Wittbrodt (iD) http://orcid.org/0000-0001-8550-7377

### Ethics

Animal experimentation: Animal experimentation: Medaka (Oryzias latipes) and zebrafish (Danio rerio) stocks were maintained as closed stocks in a fish facility built according to the local animal welfare standards (Tierschutzgesetz 111, Abs. 1, Nr. 1), and animal experiments were performed in accordance with European Union animal welfare guidelines. The facility is under the supervision of the local representative of the animal welfare agency. Fish were maintained in a constant recirculating system at 28°C with a 14 h light/10 h dark cycle (Tierschutzgesetz 111, Abs. 8 1, Nr. 1, Haltungserlaubnis AZ35-9185.64 and AZ35-9185.64/BH KIT).

### Decision letter and Author response

Decision letter https://doi.org/10.7554/eLife.32319.026

Author response https://doi.org/10.7554/eLife.32319.027

## Additional files

### Supplementary files
• Transparent reporting form
DOI: https://doi.org/10.7554/eLife.32319.024

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
