## [Decision Letter]

Thank you for submitting your article "Activating the regenerative potential of Müller glia cells in a regeneration-deficient retina" for consideration by *eLife*. Your article has been reviewed by two peer reviewers, and the evaluation has been overseen by a Reviewing Editor and Diethard Tautz as the Senior Editor. The following individuals involved in review of your submission have agreed to reveal their identity: Alejandro Sánchez Alvarado (Reviewer #1) Pamela A Raymond (Reviewer #3).

The reviewers have discussed the reviews with one another and the Reviewing Editor has drafted this decision to help you prepare a revised submission.

Summary:

This paper investigates whether medaka and zebrafish have the same capacity for retina regeneration. The authors find that in contrast to zebrafish, medaka (*Oryzias latipes*) has limited regeneration capacity. They also show that Sox2 expression is not maintained to the same extent in medaka Müller glia (MG) as in zebrafish. Forced expression of Sox2 grants medaka MG cells traits, which are more similar to injury responses observed in zebrafish. In general this is a well-executed study with highly advanced and beautiful imaging.

Essential revisions:

Reviewers agreed on the importance of this work, but also concluded that the functional assays reported are somewhat preliminary, with some of the conclusions not well supported by the reported data. As such, we would like for the authors to address these general technical concerns as follows:

1)Subsection “olMG cells reenter the cell cycle after injury but do not generate neurogenic clusters!” – The statement that medaka Müller glia are quiescent in juveniles cites a previous study (Lust et al., 2016) dealing with embryonic and larval stages, not juvenile fish. The present study also appears to use larval fish (8 days post-fertilization), as stated in the Materials and methods section. The authors need to clarify the stages being used for the reported experiments. For example, the layer of rod nuclei in the ONL in the larval medaka retina (e.g., Figure 1) is a scant one-cell thick, whereas in adult medaka retinas, there are 2-3 rows of rod nuclei (e.g. Imanishi et al., 2007,). During this early larval period, as in all species of teleost fish studied, rod photoreceptors accumulate gradually, and they are generated by rare stem cell divisions of Müller glia and the rapidly dividing rod progenitors they produce, which migrate apically into the ONL and differentiate into rods. Addressing this concern, therefore, is important.

2) Subsection “olMG nuclei but not their cell bodies are depleted after PRC injuries” and Discussion section – The statement that after photoreceptor injury medaka Müller glial "nuclei migrate apically into the wound site but remain there" and that "cell bodies are maintained at 10 days after injury in the absence of a nucleus in the INL" whereas radial glial fibers remain intact is not entirely justified. The only marker used for cell nuclei (in Figure 3) is the transgenic rx2:H2B-eGFP reporter; another possible interpretation of the results is that this reporter is no longer expressed in Müller glia that responded to the injury. This possibility is supported by the in vivo image in Figure 3, in which the arrowhead points to a Müller glial radial fiber labeled with rx2:lifeact-GFP cytoplasmic reporter. careful examination shows a 'hole' of reduced fluorescence in the soma, which likely corresponds to the nucleus in the inner nuclear layer (INL). In Figure 3, the injury site in the ONL is not apparent, and co-localization of BrdU and GFP is not readily apparent.

3) Subsection “olMG cells divide in the INL with an apico-basal distribution” – The data measuring cleavage plane of dividing Müller glia are not robust (only 6 cells were examined). In the example shown (Figure 4), there is no obvious damage in the photoreceptor layer. How do the authors exclude the possibility that these dividing cells could be neuronal progenitors in the rod lineage, reflecting normal development, not regeneration? The PH3 labeled cell (Figure 4) and the mitotic cell (Figure 4) could both represent rod progenitors. Note that the mitotic cell in Figure 4 is slightly above and in front of a Müller glial nucleus with a distinctive glial shape (elongated, polygonal), which could indicate that the mitotic cell may be a Müller-derived, early progenitor in the rod lineage.

4) Subsection “*Sox2* expression is not maintained in proliferating olMG cells after injury” – The authors find that proliferating cells in medaka do not sustain Sox 2 expression and don't form neurogenic clusters. Based on these data they assign a role to Sox2 in MG cells. Nevertheless, there are a fairly large number of Sox 2 expressing cells in medaka after injury (Figure 6). Thus, a major role of Sox2 might be sustaining proliferation in the cycling progeny of MG cells rather than only determining MG cell fate. Along the same line, it would be important to quantify Sox2 expression after the different injury models rather than just carrying out comparisons between the two species in the same injury model. More importantly: one wonders what is the percentage of Sox 2 expressing cells that are BrdU^+^? This is the counting that would support the conclusion reached rather than the percentage of BrdU^+^ cells that are Sox2^+^. This is insufficient information: how many cells were counted and from how many retinas?

5) Subsection “Sustained *Sox2* expression restores olMG driven regeneration” – Figure 7. No evidence is provided for induced expression of Sox2 in Müller glia. The effect of overexpression of Sox2 on 'regeneration' is very small, and only a single marker (HuC/D) was used to identify retinal cells. The co-localization of HuC/D and BrdU is not convincing in the absence of a panel showing HuC/D only. The quantification data are incomplete; only percentages shown, no information on number of cells counted and number of retinas. Several experiments would strengthen the conclusions in Figure 7. Please show that more MG cells express Sox 2 for example with double staining with GS. Please comment on whether MG progeny form proliferating clusters after sustained Sox 2 expression. A movie (of similar kind that is shown earlier) would be a very nice illustration to demonstrate the increased regenerative response by MG. Wildtype animals treated with mifepristone were used as control. Would be also necessary to show the control with the transgenic line without drug.

6) Discussion section – No evidence is provided for depletion of the medaka Müller glia or for "symmetric division into two differentiating cells, depleting the olMG pool". In fact, Figure 3 shows that in the region designated as a photoreceptor lesion at 2, 4, and 6 days after injury, labeling with the glial antibody glutamine synthetase (GS) shows that radial fibers of Müller glia persist at densities equivalent to the flanking, uninjured regions. These results are contrary to the conclusion that Müller glia are converted into photoreceptors by a symmetric division.

---

## [Author Response]

Essential revisions:Reviewers agreed on the importance of this work, but also concluded that the functional assays reported are somewhat preliminary, with some of the conclusions not well supported by the reported data. As such, we would like for the authors to address these general technical concerns as follows:1) Subsection “olMG cells reenter the cell cycle after injury but do not generate neurogenic clusters!” – The statement that medaka Müller glia are quiescent in juveniles cites a previous study (Lust et al., 2016) dealing with embryonic and larval stages, not juvenile fish. The present study also appears to use larval fish (8 days post-fertilization), as stated in the Materials and methods section. The authors need to clarify the stages being used for the reported experiments. For example, the layer of rod nuclei in the ONL in the larval medaka retina (e.g., Figure 1) is a scant one-cell thick, whereas in adult medaka retinas, there are 2-3 rows of rod nuclei (e.g. Imanishi et al., 2007,). During this early larval period, as in all species of teleost fish studied, rod photoreceptors accumulate gradually, and they are generated by rare stem cell divisions of Müller glia and the rapidly dividing rod progenitors they produce, which migrate apically into the ONL and differentiate into rods. Addressing this concern, therefore, is important.

We agree with the reviewers that the stages used (hatchling fish, 8dpf) need to be clarified and we have changed it accordingly throughout the text. We are now referring to the fish as hatchlings, which is the same stage we used in our previous studies.

Since the referees had pointed out that in Figure 1 the ONL of medaka is only one-cell thick and thicker in adult medaka of the mentioned study (Imanishi et al., 2007) we additionally provide now a new figure (new Figure 1—figure supplement 1) highlighting the differences in photoreceptor compositions in uninjured medaka and zebrafish both at 8dpf and in adults using immunohistochemistry for markers of rods and cones and DAPI.

While zebrafish have one layer of rod PRCs at 8dpf and at least three layers of rods in adults, medaka basically maintain the rod layer from embryonic to adult stages.

Those data clearly explain the difference in rod-production. While medaka rods are born by progenitors after the exit from the CMZ, which we show by Cre/lox-mediated lineage tracing (new Figure S2), rods in zebrafish are derived from drMG cells, explaining the difference between the two species.

We have introduced a new paragraph into the Results section highlighting this important point at the beginning of the manuscript.

2) Subsection “olMG nuclei but not their cell bodies are depleted after PRC injuries” and Discussion section – The statement that after photoreceptor injury medaka Müller glial "nuclei migrate apically into the wound site but remain there" and that "cell bodies are maintained at 10 days after injury in the absence of a nucleus in the INL" whereas radial glial fibers remain intact is not entirely justified. The only marker used for cell nuclei (in Figure 3) is the transgenic rx2:H2B-eGFP reporter; another possible interpretation of the results is that this reporter is no longer expressed in Müller glia that responded to the injury. This possibility is supported by the in vivo image in Figure 3, in which the arrowhead points to a Müller glial radial fiber labeled with rx2:lifeact-GFP cytoplasmic reporter. careful examination shows a 'hole' of reduced fluorescence in the soma, which likely corresponds to the nucleus in the inner nuclear layer (INL). In Figure 3, the injury site in the ONL is not apparent, and co-localization of BrdU and GFP is not readily apparent.

In our experience, GFP is best stabilized in a conformation where it is fused to histones.

In Figure 3 we used a double transgenic line: rx2::lifeact-eGFP, rx2::H2B-eGFP, where the same regulatory element drives H2B-eGFP as well as lifeact-eGFP.

The fact that the MG fiber is still visible indicates that rx2-driven lifeact-eGFP is still expressed. Additionally, H2B-GFP is a highly stable fluorophore, which is present long after its transcription had been downregulated.

Together these findings argue against a loss of GFP due to inactivity of the corresponding regulatory element and rather support an alternative scenario, the loss of the H2BeGFP expressing nuclei.

To make this point clearer, we adjusted the text describing these results and also the figure, clearly indicating that a double transgenic fish line had been used.

To illustrate the absence of nuclei in olMG fibers forming in response to injury we added another panel to Figure 3 (Figure 3) showing DAPI and GFP stainings at 3 dpf in rx2::lifeact-eGFP retinae.

To make the injury site and the GFP and BrdU co-localization more apparent we adjusted the former Panel 3B (now 3C) to clearly show the overlap of GFP and BrdU.

3) Subsection “olMG cells divide in the INL with an apico-basal distribution” – The data measuring cleavage plane of dividing Müller glia are not robust (only 6 cells were examined). In the example shown (Figure 4), there is no obvious damage in the photoreceptor layer. How do the authors exclude the possibility that these dividing cells could be neuronal progenitors in the rod lineage, reflecting normal development, not regeneration? The PH3 labeled cell (Figure 4) and the mitotic cell (Figure 4) could both represent rod progenitors. Note that the mitotic cell in Figure 4 is slightly above and in front of a Müller glial nucleus with a distinctive glial shape (elongated, polygonal), which could indicate that the mitotic cell may be a Müller-derived, early progenitor in the rod lineage.

We could only observe PH3 positive cells in the INL in injured fish. Given that, together with the fact that MG cells in medaka are quiescent at the stage examined and as outlined above and in contrast to zebrafish do not generate rod PRCs, we believe that these PH3-positve cells are injury-responsive MG cells.

4) Subsection “Sox2 expression is not maintained in proliferating olMG cells after injury” – The authors find that proliferating cells in medaka do not sustain Sox 2 expression and don't form neurogenic clusters. Based on these data they assign a role to Sox2 in MG cells. Nevertheless, there are a fairly large number of Sox 2 expressing cells in medaka after injury (Figure 6). Thus, a major role of Sox2 might be sustaining proliferation in the cycling progeny of MG cells rather than only determining MG cell fate. Along the same line, it would be important to quantify Sox2 expression after the different injury models rather than just carrying out comparisons between the two species in the same injury model. More importantly: one wonders what is the percentage of Sox 2 expressing cells that are BrdU^+^? This is the counting that would support the conclusion reached rather than the percentage of BrdU^+^ cells that are Sox2^+^. This is insufficient information: how many cells were counted and from how many retinas?

We followed the advice of the referees and have also quantified the number of Sox2^+^ MG cells after PRC and RGC injury, where we see a very similar trend as in the needle injuries and included this data now in the text.

With respect to the percentage of Sox2 expressing cells that are dividing cells we are a bit confused by the request of the referees. What we focus on in the manuscript is the response of olMG cells to injury and more specifically the question: is *sox2* maintained in injury responding cells?

Since in medaka only injury-responsive olMG cells are BrdU^+^, we find that it is necessary to analyze how many of these BrdU^+^ cells still express *sox2* and how many do not.

Apparently, we have not been able to state that clearly in the submitted version of the manuscript. We have therefore adjusted the text describing this experiment with a more detailed description.

Additionally, a large number of Sox2-expresing cells presented in the figure are not olMG cells but rather Amacrine cells, which also express Sox2 in homeostatic conditions and can be distinguished from MG cells due to their position and nuclear shape. We clarified those points in the revised version of the manuscript in both, text as well as in the figures.

Moreover, we have now included the numbers of cells counted and the numbers of retinae used in the figure legends.

5) Subsection “Sustained Sox2 expression restores olMG driven regeneration” – Figure 7. No evidence is provided for induced expression of Sox2 in Müller glia. The effect of overexpression of Sox2 on 'regeneration' is very small, and only a single marker (HuC/D) was used to identify retinal cells. The co-localization of HuC/D and BrdU is not convincing in the absence of a panel showing HuC/D only. The quantification data are incomplete; only percentages shown, no information on number of cells counted and number of retinas. Several experiments would strengthen the conclusions in Figure 7. Please show that more MG cells express Sox 2 for example with double staining with GS. Please comment on whether MG progeny form proliferating clusters after sustained Sox 2 expression. A movie (of similar kind that is shown earlier) would be a very nice illustration to demonstrate the increased regenerative response by MG. Wildtype animals treated with mifepristone were used as control. Would be also necessary to show the control with the transgenic line without drug.

We have addressed all of the points raised by the referees here and have extended existing figures and introduced data from new experiments.

To show that Sox2 is expressed in mifepristone induced MG cells of the transgenic rx2::Lex^PR^ OP::sox2 OP::H2B-eGFP line we performed additional experiments and have introduced a new Figure 7, which shows the increased amounts of Sox2 protein in induced olMG cells.

To address the formation of proliferating clusters after sustained Sox2 expression we have performed additional experiments.

As an alternative to the suggested life imaging over extended period of time, we now present additional data of fixed retinae, which were analyzed 3 days after injury. Sustained expression of Sox2 triggers cluster formation as an injury response. We detected the formation of BrdU^+^ clusters of proliferating cells as well as the distribution of BrdU^+^ cell intro all layers of the retina indicating the enhanced potential of the olMG cells in presence of sustained Sox2 levels. We have included this data in the new Figure 7 of the revised manuscript.

To better visualize the co-localization of BrdU and HuC/D, we have changed the previous Figure 7 (Figure 8 in the revised manuscript), so that the HuC/D label is shown alone. Additionally, we have extended the figure legends now stating all the numbers of cells counted and the numbers of retinae used.

Finally, as requested, we have now included data for the transgenic fish, which were not induced with mifepristone, but underwent the same injury, BrdU pulse and chase timeline. We see the same results as in the wildtype control fish, which were treated with mifepristone and the data is now included in the quantifications in the revised Figure 8.

6) Discussion section – No evidence is provided for depletion of the medaka Müller glia or for "symmetric division into two differentiating cells, depleting the olMG pool". In fact, Figure 3 shows that in the region designated as a photoreceptor lesion at 2, 4, and 6 days after injury, labeling with the glial antibody glutamine synthetase (GS) shows that radial fibers of Müller glia persist at densities equivalent to the flanking, uninjured regions. These results are contrary to the conclusion that Müller glia are converted into photoreceptors by a symmetric division.

We agree to the point raised by the referees and have rephrased the Discussion section of the revised manuscript accordingly. We agree that we had presented limited mechanistic insight related to the mode of division of olMG cells and have therefore omitted this part of the discussion. We have clarified the discussion of the cell bodies of olMG cells losing their nucleus in response to injury. As outlined above (Point 2) the fibers persist and we now only discuss the depletion of the olMG nucleus and the putative role of the olMG cell bodies which are likely not functionally equivalent to normal olMG cells.